# Hepatitis C Virus and the Host: A Mutual Endurance Leaving Indelible Scars in the Host’s Immunity

**DOI:** 10.3390/ijms25010268

**Published:** 2023-12-23

**Authors:** Mario U. Mondelli, Sabrina Ottolini, Barbara Oliviero, Stefania Mantovani, Antonella Cerino, Dalila Mele, Stefania Varchetta

**Affiliations:** 1Division of Clinical Immunology and Infectious Diseases, Fondazione IRCCS Policlinico San Matteo, 27100 Pavia, Italy; b.oliviero@smatteo.pv.it (B.O.); s.mantovani@smatteo.pv.it (S.M.); a.cerino@smatteo.pv.it (A.C.); d.mele@smatteo.pv.it (D.M.); s.varchetta@smatteo.pv.it (S.V.); 2Department of Internal Medicine and Therapeutics, University of Pavia, 27100 Pavia, Italy; 3Department of Molecular Medicine, University of Pavia, 27100 Pavia, Italy; sabrina.ottolini01@universitadipavia.it

**Keywords:** HCV, innate immunity, adaptive immunity

## Abstract

Hepatitis C virus (HCV) has spread worldwide, and it is responsible for potentially severe chronic liver disease and primary liver cancer. Chronic infection remains for life if not spontaneously eliminated and viral persistence profoundly impairs the efficiency of the host’s immunity. Attempts have been made to develop an effective vaccine, but efficacy trials have met with failure. The availability of highly efficacious direct-acting antivirals (DAA) has created hope for the progressive elimination of chronic HCV infections; however, this approach requires a monumental global effort. HCV elicits a prompt innate immune response in the host, characterized by a robust production of interferon-α (IFN-α), although interference in IFN-α signaling by HCV proteins may curb this effect. The late appearance of largely ineffective neutralizing antibodies and the progressive exhaustion of T cells, particularly CD8 T cells, result in the inability to eradicate the virus in most infected patients. Moreover, an HCV cure resulting from DAA treatment does not completely restore the normal immunologic homeostasis. Here, we discuss the main immunological features of immune responses to HCV and the epigenetic scars that chronic viral persistence leaves behind.

## 1. Introduction

Hepatitis C virus (HCV) is a persistent infection in most exposed individuals. After several years of a largely asymptomatic course, chronic hepatitis may develop into cirrhosis and hepatocellular carcinoma (HCC) in a proportion of patients [1]. Symptomatic acute infection is rarely observed and recovers spontaneously in about a third of cases. The mechanisms responsible for the chronically evolving infection are still poorly understood, despite the host’s immune responses to HCV having been extensively explored (reviewed in [2]). One of the reasons that may explain the aura of mystery surrounding HCV is the lack of a reliable animal model of disease that accurately reproduces the natural history of infection in humans. Indeed, this rather peculiar *Flaviviridae* family member has evolved with the human host like enemies living under one roof, ensuring pathogen persistence but, at the same time, refraining from being aggressive to secure long-term survival of the human species. In line with this view, HCV usually causes chronic low-level inflammation in the host while, at the same time, outpacing the host’s immune surveillance [3] by establishing a swarm of diverse viral variants (quasispecies) and via suppression or alterations of innate and adaptive immunity. Interestingly, HCV does not confer immunity to reinfection, and it is commonly thought that this is mainly due to virus variability, resulting in impaired recognition of variants which patients have never been exposed to. However, virus-induced long-term impairment of immune responses results in T cell exhaustion which often persists even after cure [4,5]. Moreover, the reduced production of interferon (IFN)-γ has been shown to contribute to increased susceptibility to HCV reinfection [6].

The complex strategies employed by HCV to evade detection by the immune system can lead to alterations in how immune cells function and interact, leaving behind residual effects, akin to scars, within the immune system’s memory. The advent of direct-acting antivirals (DAAs) marks a pivotal turning point in the management of HCV infection, particularly following the introduction of second-generation DAAs that target different viral proteins. This treatment regimen is well-tolerated, is of short duration (8–12 weeks), and results in eradication rates nearing 100%. However, there is still a residual risk of developing HCC [7,8]. Moreover, previous studies performed after IFN-based therapies showed a complete or partial persistence of T cell dysfunction after SVR [9,10], raising the question of whether DAA therapies can restore virus-induced immune dysfunction after years of chronic infection. Theoretically, diverse mechanisms may be triggered to reverse immune dysfunctions, including DAA-induced diminution of viral antigens and viral load [11]. However, a body of evidence indicates that not all consequences of chronic hepatitis C are completely reversible following a sustained virologic response.

The purpose of this review is to examine the immunological aspects of the HCV–host relationship and the residual impairment of selected cells involved in antiviral immunity after cure with DAAs.

## 2. HCV–Host Relationship: A Mutual Endurance

### 2.1. Innate Immunity: Friend or Foe?

Viruses, particularly those responsible for persistent infection, have a remarkable ability to become adapted to different environments. In the case of RNA viruses, such as HCV, this is in part mediated by their high mutational rates, allowing for the rapid selection of variants that can overcome hostile environments. These are initially represented by early innate defense mechanisms which are triggered immediately after infection and function to limit the extent of microbial spread. The recognition of pathogens occurs through a series of receptors that sense regular patterns of molecular structures shared by many micro-organisms but are not present on the host’s own cells. These patterns and the receptors involved in their recognition are called pathogen-associated molecular patterns (PAMPs) and pathogen recognition receptors (PRRs), respectively. The innate signaling receptors consist of a tetrad of PRRs relevant to viruses: (i) Toll-like receptors (TLRs) which sense all microbes; (ii) retinoic acid-inducible gene I (RIG-I)-like RNA helicases (RLHs) and (iii) melanoma differentiation-associated 5 (MDA-5), which both predominantly sense viruses; and (iv) nucleotide-binding oligomerization domain (NOD)-like receptors (NLRs) which sense bacteria and viruses [12]. When entering cells, viruses usually elicit a robust innate immune response, unleashing the intracellular IFN cascade leading to type I IFN production [13]. HCV is no exception since cytosolic HCV RNA induces a brisk interferon-stimulated gene response in the liver.

Upon acute HCV infection of hepatocytes, early HCV RNA recognition by PRRs induces interferon (IFN) type I (IFN-α/β) and type III (IFN-λ) gene transcription through IFN regulatory factor (IRF) 3 phosphorylation, dimerization, and nuclear translocation. Following the recognition of secreted IFNs by their cognate receptors, JAK/STAT-mediated pathways activate the expression of IFN-stimulated genes (ISGs) to induce an antiviral state. Interestingly, HCV seems to ignore early innate defense mechanisms, as it replicates almost immediately after penetration into target cells, suggesting that innate immunity is unable to control virus infection. Indeed, despite HCV flooding the hepatocytes with type I IFN, some HCV proteins interfere with IFN regulatory proteins resulting in impaired signal transmission. Indeed, NS3/4A cleaves mitochondrial antiviral signaling (MAVS) [14] and TIR domain-containing adapter-inducing IFN-β (TRIF) adapter protein [15] resulting in failure to activate IRF3 with consequent impaired activation of downstream target genes, including IFN-β. Moreover, NS3/4A protein may reduce E3 ubiquitin ligase Riplet-induced RIG-I activation [16,17] and NS3 is able to block the interaction between TBK1 and IRF3, inhibiting IRF3 activation [18]. Another mechanism exploited by the virus to inactivate IFN-β production is mediated by the NS4B protein, which blocks RIG-I-induced activation of IFN-β production through binding to stimulator of interferon genes (STING) and blocking the STING–MAVS interaction [19,20]. NS4B may also inhibit the TLR3-mediated interferon signaling pathway, inducing the degradation of TRIF [21]. Moreover, NS5A plays a role in the reduction of RIG-I- and TLR3-mediated responses [22,23]. It has been recently shown that NS5A is able to interfere with MAVS activity through binding to mitochondrial-associated protein leucine-rich pentatricopeptide repeat-containing (LRPPRC) by exploiting the ability of LRPPRC to inhibit MAVS-regulated antiviral signaling [24]. Other mechanisms of immune evasion are mediated by the HCV core protein, which can directly bind to STAT1, blocking STAT1/STAT2 heterodimerization and thus inhibiting IFN signal transduction [25,26,27]. A role in the inhibition of antiviral functions has been also reported for E2 [28] and p7 [29] proteins.

However, it is important to emphasize that the interference by HCV proteins on type I IFN signaling described above has been observed upon expression of single HCV proteins in cell lines and not in virus-infected cells, thus limiting the biological significance of such experiments.

### 2.2. T Cell Exhaustion Disrupts Adaptive Immunity and Sets the Stage for HCV Persistence

As in other virus infections, a rapid and efficient activation of the different components of the innate immune system is crucial not only for the initial containment of virus replication and spread, but also for a timely and efficient promotion of downstream adaptive responses, which require more time for their induction but are essential for eradication or at least long-term control of the infection [30]. The evidence that HCV can affect innate immune function [31], as outlined above, may have important implications with respect to priming and maturation of HCV-specific adaptive responses.

HCV-specific T cells are detectable in the liver 8–12 weeks after infection, suggesting a delayed response [32] outpaced by an impetuous HCV replication that starts only a few days after exposure [3]. The chemokines responsible for recruiting HCV-specific T cells (such as CXCR3 and CCR5) are expressed in the HCV-infected liver at an earlier stage (2–8 weeks post-infection) to prompt the intrahepatic recruitment of HCV-specific CD8+ T cells. This suggests that intrahepatic T cell responses are not delayed by late recruitment but, rather, by delayed induction of HCV-specific CD8+ T cells [33]. The importance of T cells in facilitating viral clearance is supported by the observations that self-limiting HCV infections are associated with vigorous and multi-specific T helper and cytotoxic T cell responses [34,35] and that depletion of CD8 T cells in HCV-infected chimpanzees prevents HCV eradication in this animal model [36]. This evidence is further supported by work that identified transcriptional differences between HCV-specific CD8 T cells from patients with a chronically evolving acute HCV infection and those with a self-limiting infection [37]. CD4+ T cells were found to play a key role in infection control, being required for CD8 T cell-mediated HCV clearance. However, in patients unable to clear the HCV infection, HCV-specific CD4 T cells rapidly developed proliferative defects followed by their deletion [38] in patients with persisting viremia.

Once a chronic infection is established, an important negative regulatory pathway is represented by CD4+FoxP3+ T cells that can suppress virus-specific T cells, thereby affecting the quality and intensity of the anti-viral responses [39]. Indeed, regulatory CD4+CD25+FoxP3+ T cells may contribute to T cell dysfunction in chronic hepatitis C and, indeed, blockade of Tim-3, another checkpoint molecule, on CD4+CD25+ T cells promoted the expansion of effector T cells more substantially than Tregs by improving STAT-5 signaling, thus correcting the imbalance of Foxp3+ Tregs/Foxp3− T effectors that was induced by HCV infection, in a manner similar to the PD-L1 blockade that upregulates STAT-5 phosphorylation in Tregs ex vivo [40]. The fine balance between effector and regulatory T cells would therefore be advantageous for both the pathogen and the host, allowing, on the one hand, the persistent survival of the pathogen and, at the same time, the prevention of rapidly progressive necroinflammation in the host’s liver, eventually leading to the development of severe disease and cirrhosis.

The signs of progressive CD8+ T cell exhaustion start to appear after persistent viremia is established, with viral escape mutations in CD8 epitope regions leading to the diminished recognition of the increasingly complex circulating quasispecies [41]. T cell exhaustion is typically induced by prolonged exposure to antigens during a chronic infection. The mechanisms that may reverse T cell exhaustion include a reduction in inhibitory molecules expression, such as PD1, CTLA-4, TIGIT, and TIM-3, which are known to inhibit immune responses via binding to their ligands expressed on infected cells [42]. Indeed, successful immune therapies based on the use of antibodies blocking these molecules have demonstrated the potential to reverse T cell exhaustion, and in the context of HCV infection [43,44].

### 2.3. B Cell Responses Are Not Protective but Play a Role in Lymphoproliferative Disorders

The role of antibodies in controlling HCV is far from being clarified. Antibody responses to structural and non-structural viral proteins appear 1–2 months after acute infection and their presence appears to correlate with ongoing infection rather than protection [45]. Why this occurs is still a matter of speculation but it may be due to the rapid establishment of HCV replication that outpaces B cell responses [3]. Neutralizing antibodies are certainly generated in the HCV-infected host; however, they do not seem to represent the main driving force for HCV control. This poses a serious challenge to the development of an effective traditional prophylactic vaccine and cast doubts as to the role of humoral immunity in recovery. For these reasons, the ideal objective of attaining sterilizing protective immunity remains elusive. Indeed, HCV has developed many strategies to evade humoral control [46] and although there are reasons to believe in the development of a HCV vaccine instead of pursuing difficult-to-realize elimination programs with DAAs, one attempt at assessing the protective efficacy of a vaccine has met with failure [47].

Besides a possible role of B cells in generating HCV-neutralizing antibodies, there is evidence that B cells are chronically activated in persistent HCV infections and that this phenomenon provides the pathogenetic basis for the extrahepatic manifestations, particularly lymphoproliferative disorders, arising from the chronic antigenic stimulation. The most common of such disorders is cryoglobulinemia resulting from immune complex-mediated small vessel leukocytoclastic vasculitis, a slowly progressive relapsing remitting multiorgan disease with a potentially life-threatening course. Cryoglobulins are immunoglobulins that reversibly precipitate in serum and plasma at temperatures lower than 37 °C [48]. The current evidence points to the possible role of HCV E2 envelope protein–CD81 interactions which may reduce the activation threshold of B cells, potentially leading to polyclonal and, eventually, oligo- and monoclonal expansion of B lymphocytes. Polyclonal B cell activation is a typical feature of chronic HCV infections and is associated with the upregulation of B cell activation molecules [49,50]. An in-depth analysis of B cell subsets in patients with a chronic HCV infection revealed an increased frequency of atypical memory B cells, resulting in a skewed B cell profile characterized by exhaustion markers, due to the continuous antigenic stimulation [51]. Surprisingly, however, bulk memory B cells maintain the ability to be activated in vitro upon exposure to HCV and efficiently respond to innate and adaptive immune stimuli [50,52], suggesting that chronic HCV infection does not pervasively affect B cell function.

## 3. Can Direct-Acting Antivirals (DAAs) Reverse HCV-Altered Immune Responses?

### 3.1. DAA Treatment Reduces Liver Inflammation and Fibrosis: True or False?

Upon the development of a chronic infection, the effort to control viral spread yields an exaggerated and persistent activation of antiviral mechanisms which set the stage for the development of an inflammatory environment within the liver, ultimately contributing to the onset of fibrosis. Inflammation is a protective mechanism against viral infections and tissue damage which may become detrimental in the case of persistence. During chronic HCV infections, circulating viruses trigger the activation of macrophages, including Kupffer cells, leading to inflammasome formation. This multiprotein complex senses PAMPs primarily through NLR family pyrin domain-containing 3 (NLRP3) [53] and triggers the production of pro-inflammatory cytokines, such as IL-1β and IL-18 [54,55,56,57,58]. Additionally, Kupffer cells secrete CCL5 which prompts the release of inflammatory and profibrogenic markers [59]. Thus, the activation of hepatic macrophages induces an inflammatory environment which results in the activation of quiescent hepatic stellate cells and the subsequent deposition of extracellular matrix in the liver, leading to the development of fibrosis and cirrhosis [60]. Different studies suggested that DAA treatment induces the resolution of liver inflammation, based on the observed reduction in different plasma biomarkers of inflammation, including IL18, IP-10, IL8, CCL5, IL6 [61,62]; decline in soluble CD163, a marker of macrophage activation; and reduced liver stiffness after cure [63,64,65]. Of note, DAA therapy can also lead to a reduction in ISG expression [66,67], indicating a potential restoration of the HCV-associated inflammatory environment to a more typical state. However, some studies have shown that soluble inflammatory mediators remain elevated even after successful completion of DAA treatment [68], including cases in whom treatment was initiated during the early phase of infection [69]. In addition, certain molecules, such as IL17, which were shown to be reduced during HCV infection, did not increase after DAA treatment [68]. Similar findings have been reported in cases of viral clearance following liver transplantation [70]. Notably, an inflammatory milieu was associated with HCC development in cirrhotic HCV patients after DAA treatment [71,72].

Different studies documented significant fibrosis regression following DAA therapy [73,74,75,76]. Interestingly, HCV eradication by DAAs was shown to reduce systemic oxidative stress in patients with advanced liver fibrosis [77], while others identified serum angiopoietin-2 (Ang-2) levels as a predictor of regression of liver fibrosis after successful HCV eradication by DAAs [78]. Moreover, serum CXCL10 levels predicted fibrosis regression after DAA treatment in patients with baseline levels of Mac-2-binding protein glycosylation isomer (M2BPGi) higher than the cut off index of 2 [79]. Recently, Ferreira et al. evaluated liver fibrosis before and after DAA-induced HCV clearance and found that the reduction in liver fibrosis after DAA treatment was associated with a specific TNFα and IL10 genotype. Specifically, the frequency of patients in whom an improved fibrosis stage was documented was higher in those with a TNFα 308 G/A genotype or TT for 1082 T/C and GG for 592 G/T genotypes in IL10 [80].

### 3.2. Controversial Recovery of NK Cell Function after HCV Cure

Besides type I IFN-mediated immunity, innate immune responses include several cellular lineages. Natural killer (NK) cells play an important role in the control of viral infections and have been shown to undergo significant phenotypic and functional changes at all stages of HCV infection, contributing to liver cell damage. In the acute phase of HCV infection, NK cells display an activated phenotype with enhanced cytotoxicity and IFN-γ production [81]. In chronic infection, prolonged exposure to large amounts of endogenous IFN-α produced by infected hepatocytes results in polarization toward cytotoxicity with deficient IFN-γ secretion [82]. This is caused by type I IFN-induced phosphorylation of signal transducer and activation of transcription (STAT) 1, which displaces STAT4 at the IFN-α/β receptor, resulting in decreased pSTAT4-dependent IFN-γ production and increased pSTAT1-dependent cytotoxicity [83]. The expression of tumor necrosis factor-related apoptosis-inducing ligand (TRAIL) is also upregulated in this context. The consequence of this virus-induced altered signaling is a NK cell “functional dichotomy” characterized by enhanced NK cytolytic activity and a failure to produce adequate amounts of IFN-γ and tumor necrosis factor (TNF)-α, contributing to the inability to eradicate HCV [82,84]. This profound alteration of NK cell homeostasis has been shown to be reversible upon viral eradication by treatment with DAAs [85,86,87,88,89,90], although NK cell functional recovery has not been confirmed in cirrhotic patients [91]. On the other hand, the phenotypic and functional recovery of a particular NK cell subset named the adaptive/memory subset was observed in patients with advanced fibrosis treated with DAAs [92]. The upregulation of PD-1 and reduced antibody-dependent cell-mediated cytotoxicity (ADCC) was completely rescued after viral cure [93]. Although most studies suggest an NK cell functional recovery after cure (as expected from NK cells that react promptly to virus-induced activation), there is still uncertainty as to possible residual functional defects after HCV eradication. For instance, a stochastic neighbor embedding analysis showed that HCV infection leaves an imprint on NK cells that is not reversed by DAAs [94] and down-regulation of FcγRIII, typically observed in chronic HCV infections, was not completely restored several months after DAA cure [95].

### 3.3. T Cell Exhaustion Is Only Partially Reversed after HCV Cure and Leaves Indelible Epigenetic Scars

Initial reports documented a reversal of T cell exhaustion post-DAA treatment, characterized by enhanced proliferation of HCV-specific CD8+ T cells [96], diminished PD-1 expression [97,98,99], and a transition toward a TCF-1+CD127+ memory-like T cell phenotype [100]. However, one study found that only a partial restoration of immune responses could be obtained in HCV patients treated with DAAs, with increased proliferation of HCV-specific CD4 and CD8 T cells, accompanied by an impaired ability to secrete IFNγ and IL2 [101]. Also, a recent study highlighted the importance of early DAA treatment commencement during the acute phase of HCV infection, which led to a reduction in immune exhaustion and to stronger HCV-specific T cell responses after treatment, thus reducing the risk of possible reinfection [102].

A previous study also revealed that recovery from exhaustion is related to the duration of T cell stimulation, with prolonged antigenic stimulation being associated with an irreversible state of exhaustion [103]. The authors showed that antigen removal by successful DAA treatment induced phenotypic changes in exhausted T cells, which seem differentiated toward a memory-like profile. However, these changes did not translate into a functional recovery, since the level of transcriptional regulators typical of exhausted cells, such as TOX, a key driver of T cell exhaustion, did not normalize after cure [103]. The failure of normalization of transcriptional regulators of T cell exhaustion represents an immunological scar which, intriguingly, is not detectable in T cells exposed to antigen for a limited time [103]. In line with this, another study used single-cell transcriptomics to demonstrate that, despite a decrease in terminally exhausted HCV-specific CD8 T cells after DAA, a signature of exhaustion persisted after cure [104]. Epigenetic studies analyzing the chromatin accessibility explained the reasons behind this apparently irreversible state of exhaustion. For instance, Yates et al. discovered a similar epigenetic profile in response to chronic stimulation in antigen-specific exhausted CD8 T cells in multiple human viral infections, including HCV, and showed that this epigenetic state of exhaustion signature persists for over a year after DAA-induced virus eradication [105]. Epigenetic scars included super-enhancer elements near the exhaustion-associated key transcription factors TOX and HIF1A [105]. In agreement with the above, other studies showed that checkpoint-blockade immunotherapy did not fundamentally reverse exhaustion-associated epigenetic changes, supporting the epigenetic irreversibility of exhausted T cells [106,107]. Therefore, HCV infection leaves an epigenetic signature on the host chromatin that is not fully reversed following DAA-induced virus eradication, resulting in a state of exhaustion in HCV-specific CD8 T cells that does not offer protection from subsequent viral reinfection, as demonstrated in a non-human model [108].

Notably, HCV infection influences the epigenome through alterations in DNA methylation patterns, characterized by genome-wide hypomethylation and locus-specific hypermethylation [109,110]. These changes can result in modification in gene expression, contributing to the development of chronic hepatitis C and its associated complications. HCV-induced DNA methylation changes have been observed in chronic hepatitis C infections, both with and without cirrhosis. Indeed, aberrant DNA methylation, specifically p16INK4A methylation, is frequently detected in patients with HCV-related liver diseases such as chronic hepatitis, cirrhosis, and hepatocellular carcinoma (HCC) [111]. Modifications in DNA methylation patterns can impact various biological processes and pathways, including those involved in inflammation, fibrosis, and hepatocarcinogenesis. The hypermethylation of several tumor suppressor genes leads to a downregulation of HCV-related HCC cells, resulting in the repression of protein expression [112]. The DNA methylation changes induced by HCV infection may persist after cure, potentially contributing to an increased risk of developing HCC and suggesting that viral infection causes long-term genetic and/or epigenetic damage to the liver that is not corrected after HCV cure [113]. Interestingly, among the epigenetic changes, the DNA methylation status has been associated with aging. Specifically, it has been shown that the gain or loss of CpG methylation over time is a valid method to estimate biological age [114]. An analysis of the DNA methylation status in PBMCs to calculate the epigenetic age acceleration (EAA) in chronic HCV infection showed that HCV infection induces an acceleration of epigenetic aging which was only partially reverted by DAA treatment during a long-term follow-up. Moreover, individuals who developed HCC post-eradication showed the most pronounced epigenetic aging acceleration without signs of reversal [115].

The severity of T cell exhaustion was shown to be associated with the metabolism of HCV-specific T cells, identifying enolase as a metabolic regulator of severely exhausted T cells. Thus, in chronic HCV infection, severely exhausted CD8 T cells exhibited mitochondrial impairment linked to elevated liver inflammation, and reduced enolase activity [116]. Notably, DAA therapy partially improved mitochondrial polarization in HCV-specific CD8+ T cells [116]. These findings are akin to those of others who also observed improved mitochondrial polarization in HCV-specific CD8+ T cells following viral eradication [5]. In contrast, others did not observe a mitochondrial functional recovery in HCV-specific CD8+ T cell metabolism [117].

Little data are available on the resolution of CD4 T cell perturbations after HCV cure. In one study, DAA treatment did not lead to any increase in the proliferative ability and cytokine production of HCV-specific CD4 T cells [35] while others showed a significantly increased frequency of HCV-specific CD4 T cells with reduced PD1 expression [118,119].

### 3.4. B Cell Activation May Be Reversed by DAAs even though Cryoglobulins may Persist for Years after Recovery

DAA treatment of cryoglobulinemia arising from chronic B cell activation leads to remission in approximately half of the patients, as shown in a large Italian study, which may be associated with a significant relapse rate [120]. Indeed, complete remission is reached in only about 40% of patients. Of note, the presence of pretreatment peripheral neuropathy, weakness, sicca syndrome, and renal involvement have been found to be reliable predictors of a poor clinical response in patients with HCV-related cryoglobulinemic vasculitis [121]. After HCV eradication, circulating B cell clones can persist in the absence of clinical or biochemical relapse, suggesting that these clones could have switched to a quiescent state [122]. Infections, cancer, or other endogenous or exogenous factors could trigger the proliferation of these B cells [123], in which, co-stimulation of BCR by surrogate immune complexes and of TLR9 by CpG DNA sequences has led to the clonal expansion of pathogenic quiescent B cells from HCV-cured patients with cryoglobulinemia.

The effect of DAA treatment on B cells in patients with a chronic HCV infection has been explored in a limited number of studies. T-bet+ B cells are usually expanded during chronic HCV infection and coincide with a tissue-like memory or atypical memory phenotype [51] and return to physiological levels after HCV cure after DAA therapy [124]. Similarly, HCV-specific memory B cells persisted, displayed a restored resting phenotype and normalized chemokine receptor expression, and preserved the ability to differentiate into antibody-secreting cells after cure [125].

## 4. Conclusions

In conclusion, despite DAA treatments’ ability to easily eradicate HCV, numerous unresolved questions persist. Figure 1 summarizes the phenotypic and functional features of the host’s immunity in chronic HCV infections before and after cure. First, eradicating HCV does not confer protective immunity against subsequent infections or eliminate the risk of developing hepatocellular carcinoma (HCC). Impaired immune responses that developed over years of chronic infection do not fully recover after DAA treatment. Indeed, despite the partial reversal of T cell exhaustion markers after HCV cure, there is evidence suggesting an incomplete reversal of epigenetic changes associated with T cell exhaustion. These changes persist in HCV-specific CD8 T cells after successful DAA-induced virus eradication, leaving individuals potentially vulnerable to reinfection. Furthermore, DAA treatment appears to improve Natural Killer (NK) cell functions, particularly in non-cirrhotic patients. However, a complete functional recovery of NK cells in cirrhotic patients remains uncertain, leaving certain subsets unable to fully restore their functionality post-DAA treatment. Additionally, while DAA treatment holds promise in reducing liver inflammation by potentially reversing the HCV-associated inflammatory environment, some studies indicate persistently elevated levels of soluble inflammatory mediators even after successful DAA treatment. The connection between the inflammatory milieu and the development of HCC post-DAA treatment in cirrhotic patients leaves significant concerns over the long-term prognosis in this setting.

In summary, while the beneficial effects of DAA treatment of HCV infection are undisputable, concerns persist regarding the complete recovery of innate and adaptive immunity, especially in cirrhotic patients, with possible implications on susceptibility to hepatic and extra-hepatic cancers.

## Figures and Tables

**Figure 1 ijms-25-00268-f001:**
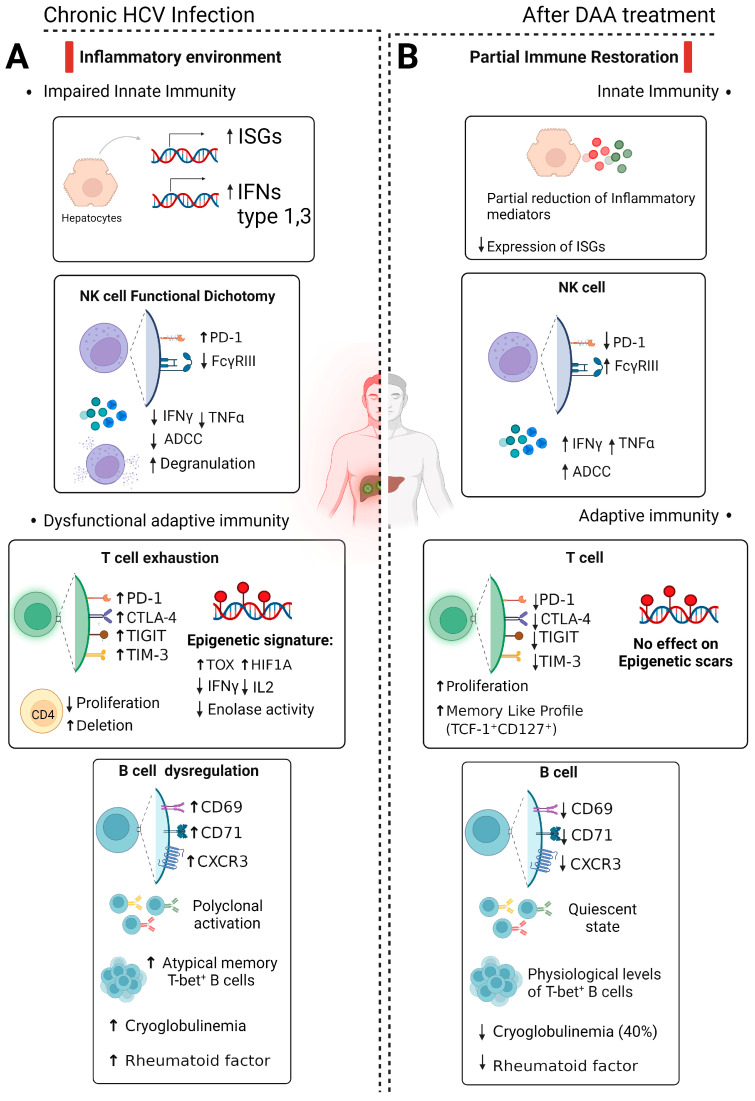
Schematic overview depicting the impact of chronic HCV infection and direct-acting antivirals (DAA) on the immune response. (**A**) During chronic infection, there is a heightened production of type I and III interferons and enhanced expression of IFN-stimulated genes (ISGs). NK cells show a functional impairment marked by increased expression of inhibitory molecule PD-1, reduced FcγRIII expression, and decreased ADCC activity and IFN-γ and TNF-α secretion, despite increased degranulation. T cell exhaustion occurs during chronic infection with increased expression of inhibitory molecules (PD-1, CTLA-4, TIGIT, TIM-3) and epigenetic imprints. B cells exhibit chronic activation, polyclonal expansion, and elevated markers of atypical memory cells. Cryoglobulinemia and rheumatoid factors may be present. (**B**) DAA treatment reduces ISG expression while maintaining a notable presence of soluble inflammatory mediators. NK cell status reverts to physiological homeostasis after viral eradication. However, complete functional restoration remains uncertain, leaving residual defects such as suboptimal antibody-dependent cell-mediated cytotoxicity (ADCC). T cell recovery post-DAA cure is also only partial, leaving long-lasting epigenetic scars. The B cell phenotype and physiological levels of T-bet^+^ B cells are restored, yet the oligoclonality of B cells may persist even after HCV eradication, potentially in a quiescent state. DAA treatment for cryoglobulinemia results in remission in only about half of the patients. Upward arrows means increase and downward arrows decrease. *Created with Biorender.com*.

## Data Availability

No new data were created.

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
