# Peer review of "Hepatitis C Virus and the Host: A Mutual Endurance Leaving Indelible Scars in the Host’s Immunity"

_ijms, 2023, doi:10.3390/ijms25010268_

Round 1

Reviewer 1 Report

Comments and Suggestions for Authors

Dear Author,

The manuscript submitted for reviewed revealed a remarkable scientific work. The authors in the current study discuss the main immunological features of immune responses to HCV and the epigenetic scars that chronic viral persistence leaves behind. However, some comments and recommendations which, when appropriately addressed, may enhance the quality of the paper.

First: Chronic HCV infection before and after HCV elimination by DAA therapy in relation to inflammatory process need to be clearer.

Second: What is the role of DNA methylation in CHC with or without cirrhosis + HCC?

Third: The author should highlight more clearly the adaptive cellular immune responses in chronic HCV infection before and after HCV elimination by DAA treatment.

Fourth: The author should add more appropriate and adequate references to related and previous work.

Fifth: The conclusions and summary are accurate but need some supported by the content.

Sixth: Minor editing of English language required. However, there are some linguistic and grammatical errors that must be rephrased and written in a correct style.

Best regards 

Comments on the Quality of English Language

Minor editing of English language required. However, there are some linguistic and grammatical errors that must be rephrased and written in a correct style.

Author Response

Thank you for providing your excellent critique which will contribute to improvement of our review article. Below is my itemized response to each comment.

One. We have significantly expanded this paragraph by adding detailed information on the mechanisms involved in liver inflammation and fibrosis and their usual return to normal homeostasis after HCV cure by DAAs (see highlighted sentences on page 5).

Two. DNA methylation has also been extensively addressed on page 7.

Three. Information and discussion on T and B cell responses have been significantly expanded by adding data on early T cell activation mechanisms and the generation of B cell proliferative disorders. The pathogenesis of mixed cryoglobulinemia has been addressed as also suggested by reviewer 2.

Four. We don't quite understand the reviewer's comment; however, we did add more references to the additional findings we have discussed.

Five. We have extended the summary and made it more factual.

Six. The manuscript has been read by a mother tongue English colleague who provided only minor edits to the manuscript for style. We have fixed a couple of typos that escaped from our attention, however, we feel that the manuscript does not need any further editing, as pointed out by reviewer 2.

Reviewer 2 Report

Comments and Suggestions for Authors

This is a comprehensive and very well-written review summarizing important aspects of antiviral immunity against HCV during natural infection and after DAA therapy. 

The manuscript covers all the important aspects of HCV-induced innate and adaptive immunity with a specific focus on the fact that HCV elimination does not fully result in a complete recovery of normal immunological homeostasis. 

I have very few comments 

:

a) Lines 117-119.  The authors correctly pointed out that “ a rapid and efficient activation of the different components of the innate immune system is crucial not only for the initial containment of virus replication and spread but also for a timely and efficient promotion of downstream adaptive responses”.

However, they do not mention the fact that T cell response is activated not early but around 4-6 weeks after infection and they didn’t comment the reason why antibodies are appearing also very late after infection. I think that a couple of lines to highlight these particular features of antiviral immunity against hepatotropic viruses ( HCV and HBV) should be mentioned.

b) The authors mention that cryoglobulinemia arising from chronic B-cell activation can be cured in approximately half of the DAA-treated patients. I think that a few words explaining what is  “cryoglobulinemia” and the molecular basis of such an event can help readers not familiar 

Author Response

We thank the reviewer for her (his) gratifying comments on our review article.

We have added comments on page 3 on the late appearance of HCV-specific T and B cells, as very appropriately suggested. In addition, we have explained our current understanding of the pathogenesis of cryoglobulinemia on pages 4-5.

Again many thanks for helping us to improve our manuscript.